# Equine Psittacosis and the Emergence of *Chlamydia psittaci* as an Equine Abortigenic Pathogen in Southeastern Australia: A Retrospective Data Analysis

**DOI:** 10.3390/ani13152443

**Published:** 2023-07-28

**Authors:** Charles El-Hage, Alistair Legione, Joanne Devlin, Kristopher Hughes, Cheryl Jenkins, James Gilkerson

**Affiliations:** 1Asia Pacific Centre for Animal Health, The Melbourne Veterinary School, The University of Melbourne, Parkville, VIC 3010, Australia; legionea@unimelb.edu.au (A.L.); devlinj@unimelb.edu.au (J.D.); jrgilk@unimelb.edu.au (J.G.); 2School of Agricultural, Environmental and Veterinary Sciences, Faculty of Science and Health, Charles Sturt University, Wagga Wagga, NSW 2650, Australia; krhughes@csu.edu.au; 3Elizabeth Macarthur Agricultural Institute, NSW Department of Primary Industries, Menangle, NSW 2568, Australia; cheryl.jenkins@dpi.nsw.gov.au

**Keywords:** equine, psittacosis, abortion, *Chlamydia psittaci*

## Abstract

**Simple Summary:**

Infectious diseases that spread from animals to humans pose a risk to both human and animal health. In recent years, the bacteria *Chlamydia psittaci* has been identified as an important cause of equine reproductive loss in Australia and has also resulted in human disease following contact with infected horses or horse tissue. This is different from the traditional pathway of transmission from birds to humans. Despite the importance of this bacteria to both horse and human health, infections in horses are incompletely understood, and risks to human and horse health remain. These risks may be exacerbated by the incomplete awareness of *Chlamydia psittaci* among Australian horse owners and others working in the equine industry. This study sought to identify the total number of cases of equine reproductive loss due to *Chlamydia psittaci* in Australia between 2018 and 2022 and thus contribute data to our growing understanding of *Chlamydia psittaci* infection in horses. A total of 31 cases were identified. These were geographically restricted to Victoria and New South Wales and were more commonly detected in the winter and spring. The results show that cases of equine reproductive loss due to *Chlamydia psittaci* remain consistent and ongoing and highlight the need for further studies.

**Abstract:**

*Chlamydia psittaci* is an important zoonotic pathogen. Although primarily a pathogen of birds, from which infection can spillover into humans and other mammalian hosts, the importance of *C. psittaci* as a cause of equine reproductive loss and the risk of infection to humans in contact with infected horses are increasingly being recognised in Australia and elsewhere. Despite the risks to both human and equine health, *C. psittaci* infection in horses is incompletely understood. This study aimed to update and summarise cases of equine psittacosis in Australia in the period 2018–2022, thus addressing a knowledge gap relating to recent cases in this country. These cases were identified from the examination of records held by state and federal veterinary authorities and from a review of published cases. A total of 31 cases were identified. Spatial and temporal trends were identified, with cases being more prevalent in winter and spring and geographically restricted to Victoria and New South Wales. The results show that cases of equine reproductive loss due to *C. psittaci* are consistent and ongoing and demonstrate the importance of routinely considering *C. psittaci* in diagnostic investigations. The need for ongoing study to better understand this important zoonotic pathogen is evident.

## 1. Introduction

Prior to 2014, *Chlamydia psittaci* (*C. psittaci*) was rarely reported as a cause of reproductive loss in horses. Since then, there have been several reports from South-Eastern (SE) Australia of late-term abortions, neonatal illness, and zoonotic spread to humans in contact with *C. psittaci*-infected horses or tissues [1,2,3,4,5]. *Chlamydia psittaci*, a species of the Chlamydiaceae family, is an obligate intracellular bacterium that is an important veterinary pathogen with zoonotic potential [6,7,8]. 

In humans, *C. psittaci* is the cause of psittacosis, a potentially fatal disease characterised by fever, malaise, myalgia, and atypical pneumonia [9,10]. *C. psittaci* is a known avian pathogen worldwide [7], and birds are considered a common source of infection for humans and other mammals. Psittacosis in humans and other mammals, including horses, results from direct or indirect interactions with infected birds or fomites, with inhalation considered the major mode of transmission [10,11,12,13]. Although transmission from other mammals to humans has only rarely been reported [14], reports over the last decade from Australia have indicated that infected horses may represent a major mammalian zoonotic threat [3,14,15]. 

Horses are considered to be occasional spillover hosts of *C. psittaci*, and infection has been sporadically associated with pneumonia, conjunctivitis, or abortion [16,17,18]. Until recently, *C. psittaci* infection has not been a primary consideration among the differential diagnoses of equine reproductive loss, with only limited epidemiological studies conducted to examine the associations between *C. psittaci* and equine abortion [1,2,5]. Throughout the last decade, *C. psittaci* has been associated with equine abortions and/or severe neonatal illness in foals in Australia, originally in the state of New South Wales (NSW) and subsequently in Victoria (VIC) and Queensland (QLD) [1,2,3,4,5]. Retrospective analysis over the last 25 years also detected *C. psittaci* from stored tissues from equine abortion cases in South Australia [19]. Given the recent emergence of *C. psittaci* as an important equine pathogen with zoonotic potential, there has been increased awareness of this disease, termed Equine Psittacosis (EP), in horses in Australia and improved surveillance in cases of equine reproductive loss and neonatal illness.

## 2. Materials and Methods

We set out to identify the total number of confirmed cases of *C. psittaci*-associated equine reproductive loss in Australia since 2018. To be included in this series of reported cases of abortion/reproductive loss, either the mare did not carry the foal to term or severe neonatal illness resulted in death within four days of birth. The diagnosis of *C. psittaci*-associated reproductive loss had to have occurred between 2018 and 2022 (inclusive) in Australia, and *C. psittaci* must have been deemed the likely cause of reproductive loss based on PCR-positive results in some or all of the following tissues/samples: foetal membranes, liver, lungs, spleen, and vaginal mucous swabs of mares, in addition to the exclusion of other known pathogens, including EHV-1 and 4 or other causative agents of equine reproductive loss [1,20,21,22]. Case numbers of equine psittacosis (EP) were determined from the records of state and federal veterinary authorities and from published reports from the years 2018–2022. In Victoria, EP cases are notifiable [23]. 

## 3. Results

From 1 January 2018 to 31 December 2022, 31 cases of *C. psittaci* associated equine reproductive loss in mares were reported to the State animal biosecurity laboratories in NSW and VIC. In total, there were 21 confirmed cases from 11 properties in NSW and 10 cases from 6 properties in VIC (Figure 1). No other Australian states or territories reported cases of *C. psittaci*-associated equine reproductive loss during the study period. Where there were multiple *C. psittaci*-associated equine reproductive loss events at a single location, cases occurred within a period of 1-3 weeks or separated by a season. One mare in late gestation located in NSW aborted a *C. psittaci* positive foal in 2019 and subsequently in 2020. Cases were restricted to the eastern areas of NSW and VIC (east of the Great Dividing Range) (Figure 2). There was no statistical difference between the average number of cases per year over the reporting period between Victoria (mean = 2.5, standard deviation (SD) = 0.58, Standard error of the mean (SEM) =1.24) and New South Wales (mean = 4.2, SD = 2.77, SEM = 0.29) (Student’s *t* test, *t* = 1.33, df = 4.33, *p* = 0.247). Nine of the cases identified here had been previously reported, including four from Victoria [1,22] and five from New South Wales [21,22].

## 4. Discussion

In this study, we have confirmed the occurrence of *C. psittaci* reproductive loss over a large geographic area within SE Australia. In two states, during each of the last 5 years, a small number of cases of *C. psittaci*-associated reproductive loss were reported. In addition, a seasonal pattern of disease was evident, with all but two cases occurring in the winter or spring (July to November in the southern hemisphere) [1,3,5,24]. This seasonal pattern has been well reported in Australian cases of EP and may be associated with cooler climatic conditions, including frost events, ecologic factors related to exposure of horses to birds and fomites, and possible changes in the shedding of *C. psittaci* by birds [5,24]. Host factors relating to the susceptibility of mares in mid- to late gestation may also explain the temporal pattern of EP. The placenta and tissues associated with late pregnancy in mares may provide a suitable environment for *C. psittaci* and inflammatory responses; however, more work needs to be carried out in this space. It should be noted that the period between infection and disease (the lag phase) remains unknown and may have implications regarding risk factors and preventative measures. Additionally, although cases have now been reported in three states following a previously reported case in Queensland before this study period [25], most *C. psittaci*-associated reproductive loss events have been centred geographically around the border between Victoria and NSW. The previously reported case in Queensland was in the south-east region close to the southern border with NSW [25], and the Victorian cases were in the north-east region approaching the northern border with NSW. One can only speculate at this stage that a combination of environmental, ecological, and management factors combine in these areas of SE Australia where larger populations of pregnant mares are exposed to psittacine birds and fomites. There remains much to tease out in terms of specific epidemiological risk factors, which may provide some explanation for this spatial and geographic prevalence of EP. 

Although outbreaks of reproductive loss and neonatal disease due to *C. psittaci* have been well reported across Australia since 2014, there is much that remains unknown about this emerging equine pathogen. This report confirms a steady prevalence in SE Australia, with a spatial and temporal trend observed. Clearly, further prospective studies are required in addition to previous epidemiologic assessments of this disease [24].

Of interest was the abortion of a mare in 2020 that had aborted due to *C. psittaci* the year before. This is suggestive of a lack of protective immunity following infection; however, it is difficult to elaborate without more details about this case. Serological evidence of *C. psittaci* infection in horses and other species has not been found to be a reliable indicator of infection [5,8,26]. This may reflect the lack of a systemic humoral response to infection and a primarily cell-mediated response to an intercellular pathogen [8]. 

*Chlamydia psittaci*-specific MLST analysis applied to the PCR-positive samples from cases in recent Australian equine cases and epizootics showed that they all belonged to the 6BC clade, ST24, and clustered with *C. psittaci* previously detected in humans, horses, and Australian parrots [27,28]. The findings from the molecular analyses are consistent with Australian native parrots being the reservoir of equine *C. psittaci* infection in Australia [5,27,28]. This is consistent with all other reported Victorian and NSW cases of EP being the parrot-associated strain. The only case outside of these two states in Australia was a pigeon-associated *C. psittaci* strain associated with abortion in a mare in SE Queensland [25]. Specific information relating to direct or indirect contact between native parrots and horses in these cases was not available, save for the fact that the majority were known to be extensively managed in areas known to have substantial psittacine bird populations. The emergence of EP in regional Australia over the last decade has reinforced serious zoonotic consequences resulting from environmental factors and interactions with wildlife and domestic species. Such spillover events involving horses, avians, bat species, and humans are jarringly familiar to equine health personnel in terms of the emergence of the Hendra virus, also prevalent in the eastern Australian seaboard [4,15,28,29]. While the emergence of EP has created widespread awareness of the disease in SE Australia, much remains to be learned about the aetiopathogenesis of what is likely a complex multispecies disease process. 

## 5. Conclusions

The emergence of EP as a cause of equine reproductive loss is now well established, and *C. psittaci* is now recognised as a zoonotic equine pathogen in Australia. This report has demonstrated that the disease should be considered endemic at a low prevalence, particularly in SE Australia. It is vital that equine practitioners, horse owners and carers, and stud personnel are aware of EP so that *C. psittaci* is a consideration in routine diagnostic testing in cases of equine abortion. Such awareness is particularly important, not only due to the threat to exposed personnel but also to facilitate accurate diagnostic processes. The emergence of such a complex pathogen, where much of the transmission pathway is complex and largely unknown, demands great scrutiny. This will enable further longitudinal studies and appropriate management strategies as more knowledge is gathered. 

## Figures and Tables

**Figure 1 animals-13-02443-f001:**
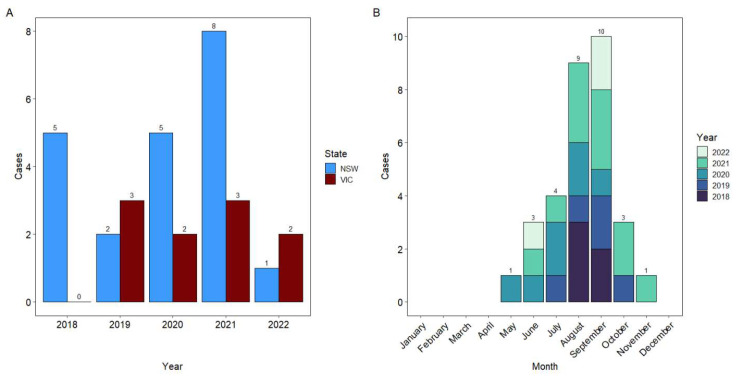
(**A**) Cases of *Chlamydia psittaci*-associated equine reproductive loss reported in Australia between 2018 and 2022, divided into state-based reporting. NSW: New South Wales, VIC: Victoria (**B**) Cases divided by the month of submission.

**Figure 2 animals-13-02443-f002:**
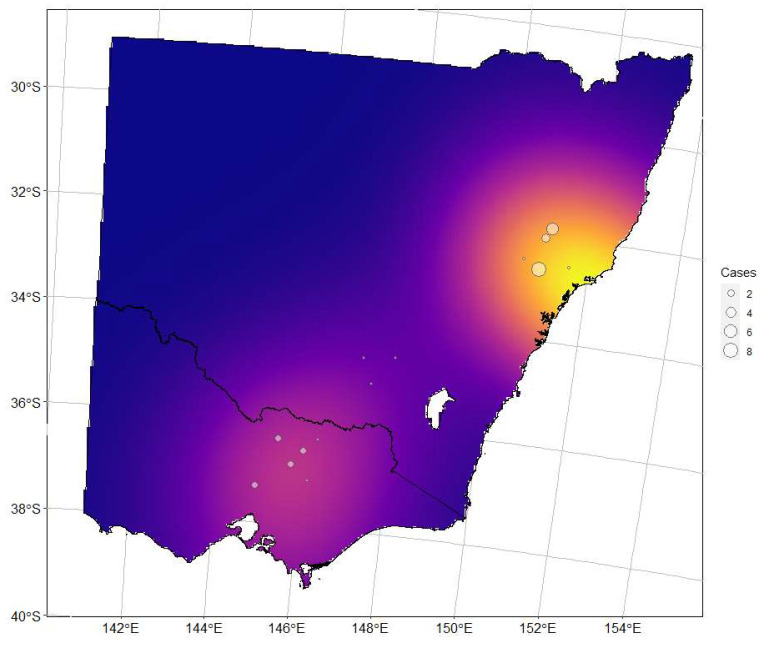
Map of Victoria (Below, South) and New South Wales (Above border line, North) with the geographic distribution of cases by postcode, with the size of the circle reflecting the number of cases within a postal code over the study period. Density kernelling is reflective of the concentration of cases in the reporting period across the two states.

## Data Availability

There were privacy constraints however data were published on the following government websites https://agriculture.vic.gov.au/support-and-resources/newsletters/vetwatch-newsletter and https://www.dpi.nsw.gov.au/.

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
