# Peer review of "Equine Psittacosis and the Emergence of Chlamydia psittaci as an Equine Abortigenic Pathogen in Southeastern Australia: A Retrospective Data Analysis"

_animals, 2023, doi:10.3390/ani13152443_

Round 1

Reviewer 1 Report

The topic is very interesting but manuscript was written very haotic. Generally it is difficult confirmed that it is kind of short communications. Authors used previous results and performed retrospective data analysis based on previously published results. Therefore it is rather review article than short communications. Range of retrospective analysis is very poor and statistical analysis and obtained results are very doubtful without appropriate statistic tools. The title requires change: e.g. Prevalence C. psittaci as emerging equine abortigenic agent in South Eastern Australia - retrospective data analysis Keywords: psittacosis is repeated two times. Materials and methods are described very general. Authors must described more details e.g. kind of PCR (methodology – if it was real-time PCR- what was target gen, method of DNA isolation e.c.)Moreover details about samples and horses should be added.

Discussion is kind of revision of available data about equine psittacosis and was written very haotic.

Author Response

Thank you for your helpful suggestions 

We have changed the title as suggested and are happy to change the nature to a review article - i have suggested this to the editor and await their response 

we have amended the materials and methods

- we feel referencing PCR methods used to confirm cases is appropriate and have made a point of referencing these molecular methods 

we have clarified many areas of the discussion and hope you may find it more suitable

we hope you can appreciate the importance of publicizing this consistent low level prevalence of a zoonotic emerging disease of horses in 2 states for this special edition. 

Reviewer 2 Report

Introduction

Line 58:  It would be helpful to the reader if there was an explanation of transmission mode of C. psittaci to horses.  Is C. psittaci transmitted to horses via infected birds?

Materials and Methods

Line 75: Suggested edit “Diagnosis of C. psittaci associated reproductive loss had to have occurred between 2018 and 2022 (inclusive)….”

Line 82: Please specify inclusive years the records for case numbers of EP were referred too.  2018-????

Results

Line 91: Suggested edit “One mare in late gestation located in NSW aborted a C. psittaci positive foal in 2019……”

Line 91:  Specify method of detection of C. psittaci in the aborted foals.

Line 95:  Is standard deviation the correct statistical method in this analysis?  Why was SD calculation selected instead of SEM?

Line 97: Please clarify nine of which cases identified here had been previously reported.  It is not clear which cases the author is referring to.  Is it nine cases in one year, or over subsequent years?

Discussion

Line 114: Comma missing between frost events and ecologic factors

Line 115:  A brief explanation of what the host factors are would be useful to the reader

Line121:  Discussion of why prevalence of C. psittaci is geographically limited to NSW and Victoria is encouraged.  Obviously limited prevalence in Qld, but no mention of South Australia?

Line 130:  Would the "of interest" case be better presented as a case study?

Conclusions

Line 159:  “It is vital that awareness of EP is…. equine practitioners and horse 159 stud personnel so that C. psittaci is a consideration in routine diagnostic testing in cases 160 of equine abortion.”  Missing words?????

A few grammatical issues noted, but easily fixed upon editing of the manuscript.

Author Response

Many thanks for your constructive suggestions please see attached document for detailed responses /corrections 

Round 2

Reviewer 1 Report

I accepted the changes in manuscript.